

# Competition and cooperation with virtual players in an exergame

Lindsay A. Shaw[1], Jude Buckley[2], Paul M. Corballis[2], Christof Lutteroth[1,3] and Burkhard C. Wuensche[1]

[1] Department of Computer Science, University of Auckland, Auckland, New Zealand
[2] School of Psychology, University of Auckland, Auckland, New Zealand
[3] Department of Computer Science, University of Bath, Bath, United Kingdom

## ABSTRACT

Two cross-sectional studies investigated the effects of competition and cooperation with virtual players on exercise performance in an immersive virtual reality (VR) cycle exergame. Study 1 examined the effects of: (1) self-competition whereby participants played the exergame while competing against a replay of their previous exergame session (Ghost condition), and (2) playing the exergame with a virtual trainer present (Trainer condition) on distance travelled and calories expended while cycling. Study 2 examined the effects of (1) competition with a virtual trainer system (Competitive condition) and (2) cooperation with a virtual trainer system (Cooperative condition). Post exergame enjoyment and motivation were also assessed.

The results of Study 1 showed that the trainer system elicited a lesser distance travelled than when playing with a ghost or on one's own. These results also showed that competing against a ghost was more enjoyable than playing on one's own or with the virtual trainer. There was no significant difference between the participants' rated enjoyment and motivation and their distance travelled or calories burned. The findings of Study 2 showed that the competitive trainer elicited a greater distance travelled and caloric expenditure, and was rated as more motivating. As in Study 1, enjoyment and motivation were not correlated with distance travelled and calories burned.

**Conclusion:** Taken together, these results demonstrate that a competitive experience in exergaming is an effective tool to elicit higher levels of exercise from the user, and can be achieved through virtual substitutes for another human player.

## INTRODUCTION

Regular exercise is instrumental to the maintenance of physical, mental and psychological health, and to achieving increased longevity (*Lee & Paffenbarger, 2000*; *Nelson et al., 2007*; *Warburton, Nicol & Bredin, 2006*). At least 150 minutes of moderate-intensity exercise, or 75 minutes of high-intensity exercise each week is recommended to attain health-related benefits (*Garber et al., 2011*). However, a significant number of people fail to initiate or maintain regular exercise at the recommended levels (*Hagströmer, Oja & Sjöström, 2007*; *American College of Sports Medicine, 1991*). Furthermore, for individuals who are prescribed exercise to address medical conditions, adherence is often low (*Jones et al., 2005*).

Corresponding author
Lindsay A. Shaw,
lsha074@aucklanduni.ac.nz

Mounting evidence suggests that exergames are a promising means for increasing physical activity in otherwise sedentary individuals (*Warburton et al., 2007*). Introducing the gameplay components of a video game to exercise has been shown to increase both exercise motivation and performance (*Peng & Crouse, 2013*; *Song et al., 2010*). Competition and cooperation are elements that appear in traditional video games and play a significant role in the enjoyment of players and their choice of games (*Vorderer, Hartmann & Klimmt, 2003*). This makes these factors ideal targets for investigation in exergaming, and past research has shown that competition and cooperation both have an influence on exercise performance, motivation, and enjoyment (*Peng & Crouse, 2013*; *Staiano, Abraham & Calvert, 2013*).

While competition and cooperation are useful features for an exergame, they have the downside of normally requiring the presence of another person. Virtual players, such as AI opponents or replays ("Ghosts") can provide a substitute for human partners. Because the behaviour of a virtual player can be controlled by the game, virtual players offer the possibility of a multiplayer experience customised to be most motivating for the player, or one that guides the player to exercise at an intensity suitable for the intended exercise outcome.

We present two exergaming systems with which we investigated the use of virtual players to provide a competitive or cooperative experience. The first is a "ghost-replay" system, in which the player is able to record play sessions and then compete against either their own recordings or the recordings of other players. In such a replay system, the user should always be motivated to improve, by focusing on beating the ghost of their last attempt. The second is an AI player in the form of a virtual "trainer" system, which adapts to the fitness level of the user. We present two user studies. The first compares the ghost replay system with a simple AI trainer. The second utilities a more advanced trainer system, based on the design of the first training system but allowing for differing behaviour profiles. This second study compares two trainer profiles: one that competes with the player and one that cooperates with them.

Using these two studies, this paper aims to answer the following research questions:

R1   How does self-competition provided by a ghost replay system influence the user's enjoyment, motivation, or exercise performance during play of a virtual reality exergame?

R2   How does playing with a competitive or cooperative trainer system influence the user's enjoyment, motivation, or exercise performance during play of a virtual reality exergame?

R3   How does the competitive inclination of the user influence the effectiveness of a competitive or cooperative trainer system on the user?

Based on existing research in this area discussed in the next section, we hypothesise that self competition via the ghost replay system should increase both the user's enjoyment and motivation in the exergame, as well as their overall exercise performance. We hypothesise

less of an effect for a trainer system than the ghost replay system, but expect that a trainer system will be more effective when aligned with the user's personality.

## RELATED WORK

There has been increasing interest in exergaming research over the last decade. The research suggests that exergames have the potential to motivate otherwise sedentary individuals to exercise (*Warburton et al., 2007*), but that they can also suffer from limited adherence similar to regular exercise (*Mestre, Dagonneau & Mercier, 2011*). Our focus here is on competitive and social factors that can hopefully improve adherence and motivate users to increase their physical activity, and on past examples of virtual training systems designed to motivate players.

Competitive and cooperative factors have been shown to influence motivation when playing exergames. *Peng & Crouse (2013)* compared three conditions in an exergame: single player versus a pre-test score, cooperation in the same physical space, and parallel competition in separate physical spaces. Their results indicate that parallel competition in separate physical spaces is particularly effective as it provides the highest enjoyment, physical activity, and motivation for future play. Interestingly, they found that cooperation was more enjoyable and motivating than solitary play, but solitary play led to greater levels of physical exertion.

Competitive factors can influence exercise performance as well as motivation. *Song et al. (2010)* looked at the effects of competitive exergame gameplay on performance and motivation in individuals with competitive and non-competitive personalities. While competition increased exercise performance in players with both personality types, non-competitive players reported lower enjoyment of the game than competitive players, and were less likely to engage in voluntary additional play. This is an important consideration for exergame design: while increased performance may offer short-term benefits, the potential long-term decrease in motivation (e.g., reduced adherence to an exercise program) for players who are not competitive likely outweighs these benefits in the long run.

Cooperative gameplay has also shown benefits in exergaming. In a study conducted by *Staiano, Abraham & Calvert (2013)*, participants played the Nintendo Wii Active game over a period of 20 weeks. Participants were assigned to either a cooperative or a competitive gameplay condition. Participants assigned to the cooperative condition lost significantly more weight than participants in the control and competitive conditions. The authors attributed the greater effectiveness of the cooperative condition to the social factors involved: as participants worked together to earn points they provided increased support and motivation for one another.

There is evidence that the use of Virtual Training systems (Virtual Trainers) influences users' motivation and exercise adherence, and may avoid some of the downsides of traditional multiplayer gaming. In particular, situations in which an individual feels stigmatised can affect exercise motivation negatively by increasing anxiety and avoidant behaviours (*Lantz, Hardy & Ainsworth, 1997*).

Current research on virtual trainers has focused on the use of a trainer separated from the gameplay. *Ijsselsteijn et al. (2006)* studied an exergame in which a virtual coach provided users with regular feedback about their heart rate. The trainer was a virtual human female character that was displayed in the corner of the screen. The feedback was provided in the form of pre-recorded voice cues and corresponding text shown in a speech bubble above the coach, e.g. "Your heart rate is too low. Cycle faster." The trainer lowered tension surrounding performance and player control, while not affecting enjoyment. The results also indicated that greater immersion in the game is linked with increased motivation.

The direct instructions used in the aforementioned study by *Ijsselsteijn et al. (2006)* have potential downsides. *Hepler, Wang & Albarracin (2012)* report that the effectiveness of these prompts and cues may rely on the personality and past behaviour of the user. For example, a user with a history of sedentary behaviour may ignore an instruction such as "cycle faster." Furthermore, the user's interpretation of feedback can have a significant effect on how it motivates the user. If the feedback is interpreted as controlling, the user may not be inclined to respond to it (*Deci & Ryan, 1985*). As a consequence, cues to exercise harder when the current level of exertion is insufficient should not be presented in a way that may be perceived as controlling, as this is likely detrimental to motivation.

*Li et al. (2014)* also examined the use of a virtual training system for active video games. In their system, the user's bodily motion was detected using a Kinect 3D sensor, and the user gained points by mimicking the motions shown on screen by the trainer. While this system had a limited degree of gamification, the research indicates that training in an immersive virtual environment is motivating.

*Wilson & Brooks (2013)* compared training with a virtual trainer in an exergame to training with a certified human trainer in a traditional exercise program. While the levels of exertion (measured by heart rate and rate of perceived exertion (RPE)) are higher with a human trainer, the results showed no significant difference in exercise adherence between the two trainer types.

In a similar study, *Feltz et al. (2014)* had participants completing exercises either alone, partnered with a human, partnered with a human-like virtual player, or partnered with a non-human-like virtual player. The partners were designed to appear more capable than the participant at the exercise task. In similar results to Wilson and Brooks, exercise performance was higher with the human partner than the virtual partners, but all partnered conditions showed higher performance than the solitary condition.

These two studies suggest that a properly designed virtual trainer could be suitable as a longer-term motivational tool for exercise. Such a trainer would also likely improve health outcomes by encouraging a greater degree of exercise performance.

While there has been a moderate amount of research on competition and cooperation in exergames, this research has been heavily focused on the use of these factors with other human players. Similarly, while there has been a moderate amount of research on virtual trainers, the training systems in existing research have little gamification and do not look at competition or cooperation.

## STUDY 1: SELF-COMPETITION VS SIMPLE TRAINER

### Methods

A cross-sectional within-subjects study was conducted to examine the effects of competition and cooperation using different representations of another player in an immersive virtual reality (VR) exergame. Specifically, the study examined exergame performance in three conditions: (1) solitary play in the exergame with no virtual player "Default Condition," (2) a ghost condition whereby participants played the exergame while competing against a replay of their performance in the first condition "Ghost condition," and (3) playing the exergame with a virtual trainer present "Trainer condition." The main outcome variables were distance travelled on the Exercycle, calories expended on the Exercycle, and RPE. In addition, the study explored participants' responses to self-report measures of enjoyment and motivation, following the completion of each condition.

A total of 22 individuals participated in the study. Three participants withdrew from the study due to suffering from discomfort related to the use of the Oculus Rift during the session. The remaining 19 (15 male, four female, mean age: 31.5, standard deviation: 9.2) were able to complete it. Informed consent was obtained from all participants, and the study was approved by the University of Auckland Human Participants Ethics Committee (reference number: 8450). The study took place between the 7th and the 21st of January, 2015.

All participants completed the control condition first in order to provide data to be used during the Ghost condition. The order of the Ghost and Trainer conditions were then randomized so as to counterbalance potential order effects.

### Design

For this research, we extended an existing exergame described in *Shaw et al. (2015)*. This exergame was chosen as it elicited high intensity exercise from the users, and was rated by the users as enjoyable. In this exergame, the user cycles along a procedurally generated track, avoiding obstacles and collecting bonuses, in an effort to maximise their score. The speed at which the user moves is governed by the rate at which they pedal on the exercycle. A 3D camera tracks their movements, allowing them to steer by leaning from side to side. The game is presented to the user via an Oculus Rift Head Mounted Display (HMD), providing them with an immersive experience.

We extended this exergame, adding a replay system and a simple virtual trainer system. The base gameplay was also slightly modified, changing obstacles to slow the player and penalize their score, rather than causing them to replay a section. This was necessary in order to avoid divergence between the ghost replay system described below, and the user's current play session.

Figure 1 shows a screenshot of the exergame, and illustrates some of the gameplay elements.

The exergame allows for playback of a participant's previous attempts through a "ghost racer" system, in which the participant sees a non-interactive replay of the past attempt on

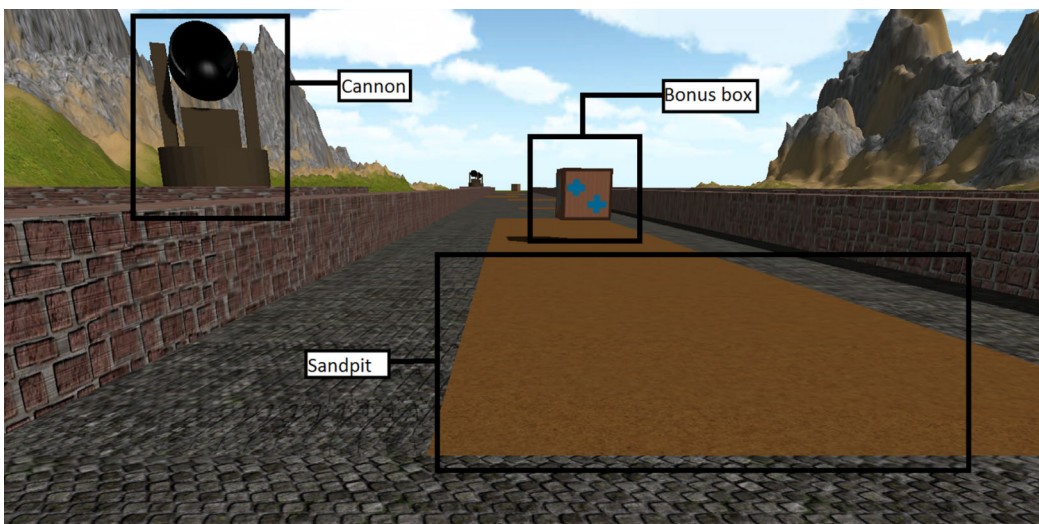

**Figure 1** Sample screenshot of the exergame showing cannons, sandpits, and a bonus on the track.

the track as they play. This offers encouragement to exercise harder in order to beat their previous attempt. When users are lagging behind their "ghost," they are also able to see points where their previous run failed to avoid obstacles, and thus they may be able to adapt their play to avoid more obstacles.

The player's ghost has the same appearance as the trainers (shown in Fig. 2): a simplified figure on a bike. When close to the player, the ghost and trainers are semi-transparent, increasing in opacity as they move further away. This is to prevent them from blocking the players view of obstacles or bonuses and becoming a potential source of frustration.

The simple virtual trainer system behaves in a similar fashion to the ghost system. When the player begins to move, another 'player' appears and travels along the track with them. However, rather than showing previous behaviour, the trainer attempts to show optimal behaviour, both in terms of gameplay and exercise. With regard to gameplay, the trainer chooses an optimal path through the track, avoiding all obstacles. With regard to exercise, the trainer adjusts its speed to guide the user towards an ideal exercise heart rate, as explained below.

First, the current heart rate of the user, as measured by the handlebar sensors, is used to estimate a relative heart rate, i.e., a percentage of the user's expected maximum heart rate based on their age. This is done using *Tanaka, Monahan & Seals (2001)* regression equation $208 - 0.7 \times age$. The trainer attempts to set a speed suitable for keeping the user's heart rate at the level associated with moderate to vigorous exertion, i.e., 64–90% of their expected maximum heart rate (*Garber et al., 2011*). If the user's heart rate is below 64% ("low heart rate"), the trainer increases its speed, requiring the user to work harder to catch up. If their heart rate exceeds 90% of their maximum ("high heart rate"), it decreases its speed, allowing them to exert less effort to keep pace. While the user's heart rate is in the target zone ("average heart rate"), the trainer stays a short distance in front of the user providing a target to follow in order to motivate them.

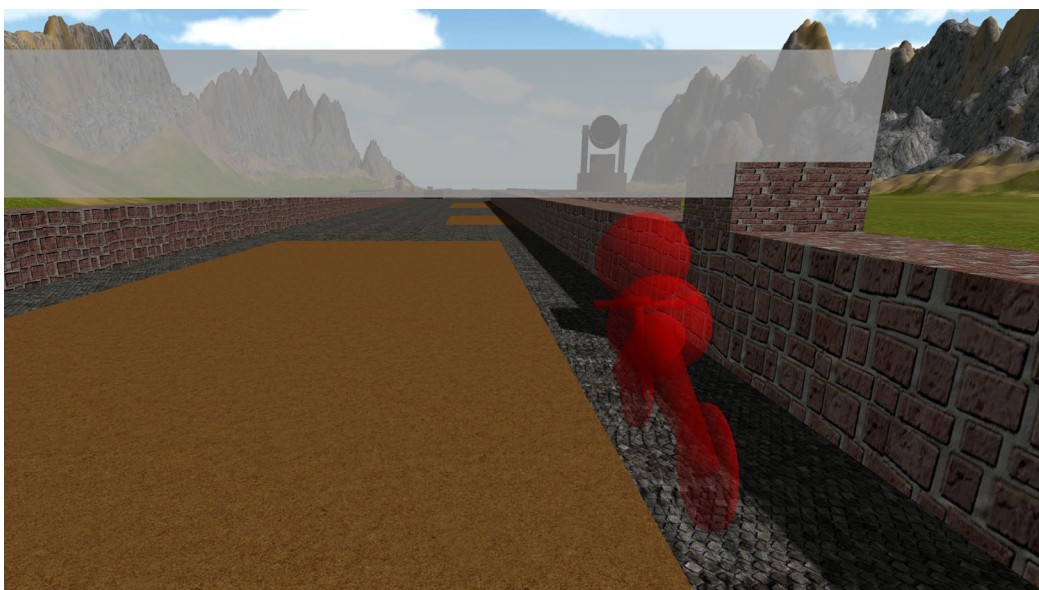

**Figure 2 Screenshot of the exergame showing the trainer in front of the player.** The red colour indicates that the player's heart rate is in the high zone. Visible at the top of the image is an overhead beam, which the trainer is ducking underneath.

## Procedure

Participants completed a pre-test questionnaire to provide general demographic data: their age, gender, and baseline self-report measures of the typical number of hours spent exercising and playing video games each week. Participants were then given a written outline of the test procedure and conditions, and written instructions on how to play the exergame. They were assisted with adjusting the exercycle and motion tracking equipment.

Participants first completed the ten-minute "Control" condition, followed by the "Ghost" and "Trainer" conditions in a counterbalanced order, separated by five-minute breaks. In the control condition, the participant's play attempt was recorded to provide the ghost for the later Ghost condition. During the break following each condition, participants were given a sheet listing the different exertion levels on the Borg RPE scale (*Borg, 1982*), and asked to rate their level of exertion. They were also given a post-condition questionnaire in which they were asked to rate how enjoyable and motivating they found the condition, and were invited to give feedback about the Ghost and Trainer systems, and about the exergame in general.

## Measures
### Distance

For each condition, the distance travelled in kilometers on the exercycle was assessed as the total kilometers travelled at the end of each exercise session. This was measured from the exercycle's output.

### Calories expended

The total kilocalories expended on the exercycle as the total Calories expended at the end of each exercise session. This was measured from the exercycle's output.

### Rate of perceived exertion (RPE)

The RPE scale (*Borg, 1982*) is a brief self-administered rating scale that was designed to measure an individual's subjective rating of exercise intensity. At the end of each exercise session, participants rated their perception of effort or "how hard they felt they had worked" during each exercise session, using a scale ranging from "6" (least exertion) through to "20" (most exertion).

### Enjoyment

Enjoyment was assessed with one item. Participants were asked to rate the statement "I enjoyed playing the exergame" on a seven-point Likert rating scale. Ratings ranged from 1 (Strongly disagree) to 7 (Strongly agree).

### Motivation

Motivation was assessed with one item. Participants were asked to rate the statement "I found the exergame motivating" on a seven-point Likert rating scale. Ratings ranged from 1 (Strongly disagree) to 7 (Strongly agree).

## Data analyses

The normality and sphericity assumptions of repeated measures analysis of variance (RM-ANOVA) were tested with the Shapiro-Wilk test and Mauchly's sphericity test, respectively. With a p-value threshold of 0.05, the normality assumption holds for the measures of distance travelled, calories expended, and RPE, but does not hold for the measures of enjoyment and motivation. The sphericity assumption holds for all measures except motivation.

RM-ANOVA with post-hoc Bonferroni tests were conducted to examine the effects of the Control, Ghost, and Trainer conditions on distances travelled, calories expended, and RPE.

Due to the non-normally distributed data, the effects of the three conditions on enjoyment and motivation were examined with Friedman tests.

Pearson correlation analyses were used to examine the association between the participant information gathered during the pre-test, and the measures listed above.

## Results and discussion

Table 1 shows the means and standard deviations of the various measures across the three conditions.

The results of a RM-ANOVA showed that there is a significant main effect Condition on distance travelled ($F = 15.28$, $p < 0.001$). The results of the post-hoc Bonferroni tests showed that distance travelled in the Trainer condition was significantly lower than in the Control condition ($p = 0.001$), and the Ghost condition ($p < 0.001$). There was no significant difference between the Control and Ghost conditions.

**Table 1 Summary of distance traveled (km), calories burned, rate of perceived exertion (RPE), and rated enjoyment and motivation in Study 1.**

|  | Control | | Ghost | | Trainer | |
|---|---|---|---|---|---|---|
|  | **Mean** | **SD** | **Mean** | **SD** | **Mean** | **SD** |
| Distance | 3.98[a] | 0.73 | 4.02[a] | 0.88 | 3.57[b] | 0.79 |
| Calories | 78.74[a] | 11.57 | 79.16[a] | 15.32 | 73.63[a] | 14.16 |
| RPE | 15.74[b] | 1.29 | 16.79[a] | 1.62 | 16.37[b] | 1.89 |
| Enjoyment | 5.58[b] | 1.31 | 6.16[a] | 1.21 | 5.37[b] | 1.68 |
| Motivation | 5.68[a] | 1.49 | 6.11[a] | 1.15 | 5.68[a] | 1.53 |

**Note:**

N = 19. For each outcome assessment, means sharing a letter in their superscript (a, b) are not significantly different at the 0.05 level. Significant mean differences at the 0.5 level are indicated by a > b.

There was also a significant main effect Condition on calories expended (F = 4.64, p = 0.16). However, the results of the post-hoc Bonferroni tests did not show any pairwise significances.

There was a significant main effect of Condition on RPE (F = 3.79, p = 0.032). The results of the post-hoc Bonferroni tests showed that the RPE rating for the Ghost condition is significantly higher than for the Control condition (p = 0.01). There were no significant difference between the Trainer condition and either of the other two conditions.

The results of a Friedman test showed a significant main effect Condition on enjoyment across the three conditions (p = 0.016). The Ghost condition was significantly more enjoyable than the Control and Trainer conditions.

The results of a Friedman test showed no significant main effect Condition on motivation across the three conditions (p = 0.370).

The results of a Pearson correlation showed that enjoyment of a condition, and level of motivation in that condition have no significant correlation with distance travelled in the condition.

The use of player recordings of past performance to encourage self-competition shows significant promise to encourage users to exercise via an exergame, particularly if they enjoy competition. Verbal and qualitative feedback from the participants indicated that being able to see and beat their previous attempt was highly enjoyable during the Ghost condition. This study failed to show benefits for the use of a multiplayer-style virtual trainer system, however that may be due to flaws in the trainer system discussed further below.

It is not too surprising to see that the Ghost condition did not encourage players to exercise significantly harder than in the Control condition. Unless turning around to look directly backwards, players would only see their ghost when it was ahead. Thus they would only receive motivation from the ghost to speed up when doing worse than it.

Attitudes towards the trainer system were less positive. Overall, it was not significantly more enjoyable than playing in the absence of another player. From verbal and written feedback, several of the participants who did not find the trainer system motivating found that it reduced their enjoyment of the exergame, citing unrealistic behaviour: "The trainer

system moved strange." We suspect this may be related to the framing of the trainer system. While the ghost system was clearly competitive, the trainer system had no particular framing as either competitive or cooperative. If the trainer was ahead, it would show participants an optimal performance, but participants were able to push themselves above the target heart rate and pull ahead; "competing" with it.

The trainer system was extremely effective at avoiding obstacles, and often navigated through obstacles with superhuman dexterity. When this occurred, participants tended to react negatively, stating that they felt that the trainer was "cheating," and was not helping them as it was not showing them an optimal path that they were capable of following.

However, it should be noted that while the Ghost condition was better received than both the Control and Trainer conditions, the overall participant response to all three of the conditions was generally positive, with the mean enjoyability and motivation ratings still being high. The exergame in general was regarded as enjoyable and motivating, and the Trainer system did not detract from that.

## STUDY 2: COMPETITIVE VS COOPERATIVE TRAINER

### Methods

A cross-sectional within-subjects study was conducted to examine the effects of competition and cooperation with a virtual trainer in the exergame environment. The effects of: (1) solitary play in the exergame with no virtual player (Default Condition), (2) competition whereby participants played the exergame while competing against a virtual trainer with a competitive behaviour profile, and (3) cooperation whereby participants played the exergame working with a virtual trainer with a cooperative behaviour profile on distance travelled and calories expended while cycling. Participants were recruited through open advertisement. A total of 28 individuals participated in the study, of which 25 (21 male, four female, mean age: 24.3, standard deviation: 9.2) were able to complete it. Informed consent was obtained from all participants, and the study was approved by the University of Auckland Human Participants Ethics Committee (reference number: 8450). The study took place between the 20th of July and the 21st of September, 2015.

### Design

The virtual trainer system described in Study 1 was fairly limited in its capability to interact with the user. For the second study, we designed and implemented a more advanced virtual trainer (shown in Fig. 2) based on the evaluation of the initial trainer design. The full design of the advanced trainer system is detailed in *Shaw et al. (2016)*.

As research discussed earlier in this paper indicates, competition as part of an exergame can affect different users very differently depending on how competitive they are. The behaviour of the first trainer system was not clearly framed as either competitive or cooperative. The advanced trainer system was designed to be customizable for either competition or cooperation in order to appeal to different personality types. In order to

do that, the advanced trainer implements two behaviour profiles: a competitive profile and a cooperative one. While the competitive trainer profile is programmed to challenge and race against the player, the cooperative trainer profile attempts to help the player achieve a higher score.

Similar to the previous trainer, the advanced trainer always chooses a path that is close to optimal for scoring points and attempts to avoid obstacles. The trainer looks ahead to avoid obstacles in the distance. For example, if there is an obstacle in the centre of the track, and beyond that is one on the left side, the trainer will choose to go right when avoiding the central obstacle. As such, a user can follow the trainer and potentially achieve a higher score. However, the trainer only looks 40 m ahead when planning its path. Beyond this point the flat nature of the track (visible in Figs. 1 and 2) and resolution of the Oculus Rift make it difficult for a human to clearly make out obstacles. Limiting the trainer's perception to this distance helps to keep its navigational ability on par with that of a human.

A moderate number of participants in Study 1 (6 of 19) mentioned the unrealistic agility of the first trainer system as something that they did not like about it. For this reason, the lateral movement speed of the advanced trainer was capped at the maximum achievable via the motion controls used by the human player.

### Competitive trainer

The advanced trainer modifies its behaviour based on the heart rate of the user, considering the same low, average, and high heart rate zones as the simple trainer (see Fig. 3). For the competitive trainer profile, when the player's heart rate is too low, the trainer's speed will increase up to 1.3 times that of the player. When in the average heart rate zone, the trainer's speed will approximately match that of the player. And when the player is in the high heart rate zone, the trainer's speed will drop down to 0.7 times that of the player.

The speed of the trainer also takes into consideration the distance from the player. If the player spends an extended time in the low heart rate zone, it is important that the trainer does not get too far ahead, otherwise it may be demotivating, or at least no longer motivating if the trainer has moved out of the player's view. Similarly, it is important that the trainer does not fall too far behind if the player is spending an extended period of time above the target heart rate zone, otherwise the benefit of the trainer will be lost even if the player's heart rate falls back into the target zone. Because the game is presented via HMD, clamping the trainer's distance means that the player is always able to look over their shoulder and see the trainer following them.

While in the target zone, the speed variation means that the trainer behaves as a human player of similar abilities, in that it occasionally pulls slightly ahead and occasionally falls slightly behind. This means that the user is always being made aware of the presence of the trainer and is encouraged to compete and stay ahead. As the user tends towards the upper end of the target heart rate zone, the trainer spends more time behind the user, while at the lower end of the zone it spends more time ahead of the user.

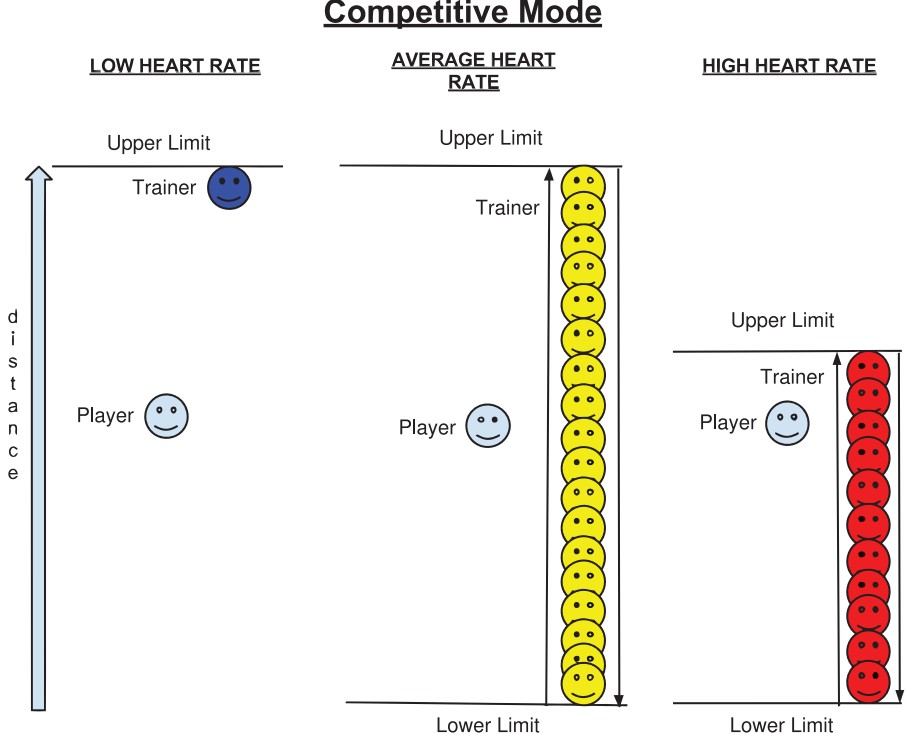

**Figure 3** Colour and position of the competitive trainer relative to the player.

### Cooperative trainer

The cooperative trainer is designed to cooperate with the player, providing assistance to the player in achieving the goals of maximising their score and maintaining an ideal target heart-rate. The cooperative trainer always gives the player a target to focus on, much like a lead cyclist in real-world group cycling activities. Following the cooperative trainer helps a player to follow a near optimal path along the track, avoiding all obstacles. Within this game, cooperation is one-directional: the trainer cooperates with the player, but the player is not providing meaningful assistance to the trainer.

The cooperative trainer uses a similar heart rate based mechanism to the competitive trainer, but it always sits in front of the player, regardless of speed. If the user is in the low heart rate zone, the trainer maintains a position well ahead of but clearly visible to the player. In the high heart rate zone the trainer stays only barely ahead of the player. And in the average heart rate zone, the trainer varies its position in a similar fashion to the competitive trainer, but within the bounds given by its positions when the player is in the high or low heart rate zones (see Fig. 4).

### Procedure

Participants completed a pre-test questionnaire to provide general demographic data: their age, gender, and baseline self-report measures of the typical number of hours spent exercising and playing video games each week. As part of the pre-experiment questionnaire, participants also filled out the Sport Orientation Questionnaire (SOQ)

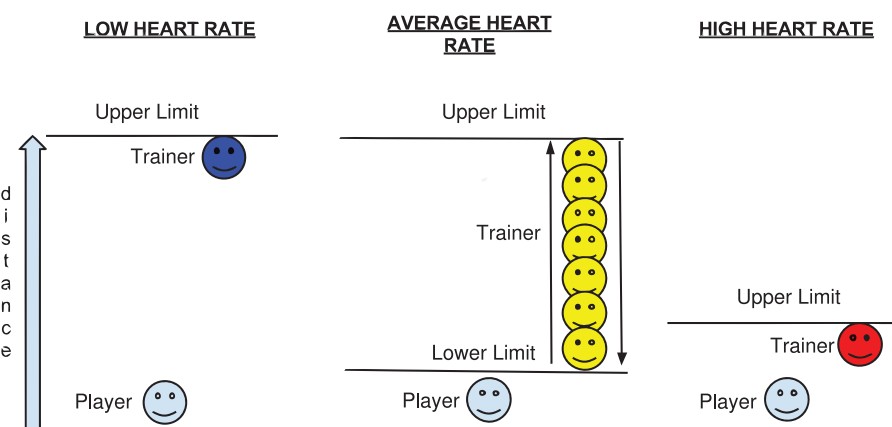

**Figure 4** Colour and position of the cooperative trainer relative to the player.

(*Gill & Deeter, 1988*) and the Task and Ego Orientation in Sport Questionnaire (TEOSQ) (*Duda, 1989*). These are validated and commonly used questionnaires that provide five personality metrics related to competitiveness in sporting activities: competitiveness, goal orientation, and winning orientation from the SOQ, and task and ego orientation from the TEOSQ.

Following the questionnaires, participants were then given a written outline of the test procedure and conditions, and written instructions on how to play the exergame. The investigator assisted participants with adjusting the exercycle and calibrated the motion-tracking equipment.

Participants completed the three conditions in a counterbalanced order determined by the method of Latin Squares. Prior to each condition, the participants were given a verbal explanation of that condition, including an explanation of the trainer's behaviour. The three conditions were separated by five-minute breaks. During the break following each condition, participants were given a post-condition questionnaire in which they were asked to rate how enjoyable and motivating they found the condition, and were invited to give general feedback about the trainer (where applicable) and the exergame in general.

## Measures
### Distance
For each condition, the distance travelled in kilometers on the exercycle was assessed as the total kilometers travelled at the end of each exercise session. This was measured from the exercycle's output.

### Calories expended
The total kilocalories expended on the exercycle as the total Calories expended at the end of each exercise session. This was measured from the exercycle's output.

**Table 2 Summary of distance traveled (km), calories burned, enjoyment, and motivation in Study 2.**

|  | Control | | Competitve | | Cooperative | |
|---|---|---|---|---|---|---|
|  | **Mean** | **SD** | **Mean** | **SD** | **Mean** | **SD** |
| Distance | 4.22[b] | 0.66 | 4.46[a] | 0.76 | 4.06[b] | 0.67 |
| Calories | 64.88[b] | 6.71 | 67.96[a] | 8.59 | 63.92[b] | 7.57 |
| Enjoyment | 5.52[a] | 1.00 | 5.24[a] | 1.69 | 5.56[a] | 1.42 |
| Motivation | 5.00[b] | 1.35 | 6.16[a] | 0.75 | 5.12[b] | 1.76 |

**Note:**

N = 25. For each outcome assessment, means sharing a letter in their superscript (a, b) are not significantly different at the 0.05 level. Significant mean differences at the 0.5 level are indicated by a > b.

### Enjoyment

Enjoyment was assessed with one item. Participants were asked to rate the statement "I enjoyed playing the exergame" on a seven-point Likert rating scale. Ratings ranged from 1 (Strongly disagree) to 7 (Strongly agree).

### Motivation

Motivation was assessed with one item. Participants were asked to rate the statement "I found the exergame motivating" on a seven-point Likert rating scale. Ratings ranged from 1 (Strongly disagree) to 7 (Strongly agree).

## Data analyses

The normality and sphericity assumptions of RM-ANOVA were tested with the Shapiro-Wilk test and Mauchly's sphericity test, respectively. With a p-value threshold of 0.05, the normality assumption holds for the distance and calories measures, but does not hold for the enjoyment and motivation measures. The sphericity assumption holds for all measures except motivation.

RM-ANOVA with post-hoc Bonferroni tests were conducted to examine the effects of the Control, Competitive, and Cooperative conditions on distances travelled and calories expended.

Due to the non-normally distributed data, the effects of the three conditions on enjoyment and motivation were examined with Friedman tests.

Pearson correlation analyses were used to examine the association between the participant information gathered pre-test, and the measures listed above.

## Results and discussion

Table 2 shows the means and standard deviations of the various measures across the three conditions.

The results of a RM-ANOVA showed that there was a significant main effect Condition on distance travelled (F = 7.4, p = 0.002). The results of the post-hoc Bonferroni-corrected tests showed that distance travelled in the Competitive condition was significantly higher than in the Control condition (p = 0.047), and the Cooperative condition (p = 0.006). There was no significant difference between the Control and Cooperative conditions.

There was a significant main effect Condition on calories expended (F = 6.63, p = 0.003). The results of the post-hoc Bonferroni tests showed that calorific expenditure in the Competitive condition was significantly higher than in the Control condition (p = 0.022), and the Cooperative condition (p = 0.011). There was no significant difference between the Control and Cooperative conditions.

The results of a Friedman test did not show a significant difference in enjoyment across the three conditions (p = 0.756).

The results of a Friedman test showed a significant difference in motivation across the three conditions (p = 0.027). The Competitive condition was significantly more motivating than the Control and Cooperative conditions.

Enjoyment of a condition, and level of motivation in that condition showed no significant correlation with distance travelled in the condition.

Task orientation as measured by the TEOSQ does not show any significant correlation with any other measurement.

Ego orientation as measured by the TEOSQ shows a moderate positive correlation with distance travelled in the baseline condition (r = 0.41, p < 0.05), but no significant correlation with distance in the other conditions, or with calories burned in any of the conditions. It shows a moderate negative correlation with motivation in the cooperative condition (r = −0.41, p < 0.05), but no significant correlation with motivation in the other two conditions. There was no significant correlation between ego orientation and enjoyment of any of the conditions.

None of the SOQ measurements showed any significant correlation with distance travelled or calories burned in any of the conditions. They also failed to show any significant correlation with reported enjoyment or motivation in any of the conditions.

Time spent on regular exercise and time spent playing video games do not appear to be correlated with the personality metrics measured by the SOQ and TEOSQ. No significant correlations are shown between these lifestyle factors and the personality traits. Surprisingly, for the participants in our study, there was also no significant correlation between time spent exercising and exercise performance in the exergame (distance travelled and calories burned measures). However, time spent on regular exercise did show a moderate negative correlation with rated enjoyment of the baseline condition (r = −0.41, p < 0.05).

Unsurprisingly, enjoyment and motivation for each of the conditions were closely related, with a moderate positive correlation between enjoyment and motivation of the default condition (r = 0.49, p < 0.05), and the competitive condition (r = 0.46, p < 0.05). This was most noticeable in the cooperative condition, with a strong positive correlation between enjoyment and motivation (r = 0.77, p < 0.001).

The competitive trainer provided a fairly interactive experience for the participants. Regardless of whether they were ahead of the trainer or behind it, the variations in the trainer's speed caused the race to always appear to be in a situation where the lead could be taken by either player. The cooperative trainer however, seemed less interactive. While its behaviour was more strictly defined as cooperative than that of the trainer in Study 1, simply receiving assistance the trainer is an experience of limited interactivity.

This is reflected in our results with regard to distance travelled, calories burned, and the motivation rating.

It is interesting to note that the cooperative trainer did not prove to be particularly more effective for non-competitive individuals. This may be because overall, it provided only limited cooperative gameplay elements. Cooperation with the trainer is one directional. While the trainer can help the player by showing them an ideal path and giving them a target to focus on, the player cannot affect the trainer beyond their heart rate changing its speed. Thus players can get the feeling of being helped, but not of helping. The moderate negative correlation between ego orientation and motivation when playing with the cooperative trainer is interesting. In this case, it may be because the non-ego oriented players found receiving assistance with their gameplay to be motivating.

Our results show an interesting contrast with the findings of *Song et al. (2010)*. Like their results, our results do not show non-competitive individuals performing worse in the competitive condition. However, their results showed reduced enjoyment and motivation for non-competitive individuals in a competitive experience. Our results, however, do not show that. This may be because in our case, the competition is against a virtual trainer, rather than another human. Further, in our study the player's opponent is designed to match their fitness level, thus the competition is always fair. These factors likely reduce the negative aspects of a competitive experience for non-competitive individuals.

Verbal and written feedback about the trainers' behaviour was consistent with that given in Study 1: when the trainers avoided obstacles in a fashion that the participants perceived as inhuman, the participants reacted negatively. Despite the fact that the agility of the trainer was reduced to a human-like level in Study 2, a Chi-Square test showed no significant difference in the number of participants mentioning "unrealistic" or "cheating" movements in their open feedback (Study 1 N: 6, Study 2 N: 9, p = 0.76). Player perceptions of the behaviour of virtual trainers and players would be an interesting future research area.

## LIMITATIONS

These two studies suffer from some limitations in their experimental design and procedure. In Study 1, the need for a dataset to be used by the ghost replay system meant that the default condition could not be counterbalanced with the other two conditions. Additionally, as mentioned above if a participant was beating their ghost, they would have to look behind in order to see it and compare their performance. However, if they pulled too far ahead the ghost could end up too distant to see.

In Study 2, the user's RPE was not measured. As such, we are unable to see how this is influenced by the competitive or cooperative trainers.

Our use of the SOQ and TEOSQ in Study 2 to analyse the participant's personality may be a limiting factor when considering the effectiveness of our trainer systems. The SOQ and TEOSQ are designed to measure personality traits in a sporting context. While an individual may be generally competitive, it is not unreasonable to assume that they could be competitive in a sporting context but not when playing video games, or vice versa.

Furthermore, the physical component of our exergame: cycling on an exercycle, is not itself a sporting activity.

In both studies, participants were experiencing a novel technology: the Oculus Rift HMD. It is possible that some of the participants in the study chose to participate in order to access this technology, and this may have influenced their perception towards the exergame. As consumer grade HMDs become more available, the risk of this will hopefully decrease for future work.

Both studies recruited participants through open advertisement, and in both cases a larger number of males than females responded. This may be due to greater male interest in playing an exergame, which is consistent with the fact that more males than females play video games (*Lucas & Sherry, 2004*; *Williams, Yee & Caplan, 2008*). As such, the findings of these studies may be biased towards individuals inclined to play video games.

In both studies, the "enjoyment" and "motivation" constructs were only measured with a single item in the post-condition questionnaires. This reduces their reliability in assessing the opinions of the participants.

## CONCLUSIONS

We have presented a set of systems for an immersive VR exergame that attempt to provide the benefits of a multiplayer experience with regard to the use of competition and cooperation as a motivational tool.

Our results indicate that competition is a useful tool in exergaming, but do not show that, i.e. necessarily the case for cooperation. Virtual players, either a replay or an AI trainer provide an effective substitute for a human player in order to increase the motivation of the user, and can increase the user's exercise performance. However a cooperative virtual player appears no more effective than solitary play.

Interestingly, our results do not indicate an influence for the personality of the player on what kind of virtual trainer system they prefer.

Using the user's heart rate as a tool for governing the behaviour of a virtual trainer appears an effective means of balancing the trainer's performance such that the user exercises at a worthwhile intensity.

There are two main implications that our results hold for the design of virtual players for use in exergaming systems. Firstly, our studies indicate that a more interactive experience leads to greater exercise intensity, likely through greater player investment in the experience. Secondly, the experience should be clearly competitive or cooperative as an experience with unclear orientation may be less effective than solitary play.

### Funding

This work was supported by the University of Auckland Faculty Research Development Fund [grant number 3709146]. The funders had no role in study design, data collection and analysis, decision to publish, or preparation of the manuscript.

## Grant Disclosures

The following grant information was disclosed by the authors:

University of Auckland Faculty Research Development Fund: 3709146.

## Competing Interests

The authors declare that they have no competing interests.

## Author Contributions

- Lindsay A. Shaw conceived and designed the experiments, performed the experiments, analyzed the data, wrote the paper, prepared figures and/or tables, performed the computation work, reviewed drafts of the paper.
- Jude Buckley analyzed the data, wrote the paper, reviewed drafts of the paper.
- Paul M. Corballis reviewed drafts of the paper.
- Christof Lutteroth conceived and designed the experiments, analyzed the data, contributed reagents/materials/analysis tools, performed the computation work, reviewed drafts of the paper.
- Burkhard C. Wuensche conceived and designed the experiments, contributed reagents/materials/analysis tools, performed the computation work, reviewed drafts of the paper.

## Ethics

The following information was supplied relating to ethical approvals (i.e., approving body and any reference numbers):

University of Auckland Human Participants Ethics Committee Reference number: 8450.

## Data Deposition

The raw data has been supplied as Supplemental Dataset Files.

## Supplemental Information

Supplemental information for this article can be found online at http://dx.doi.org/10.7717/peerj-cs.92#supplemental-information.

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
