# Peer review of "Competition and cooperation with virtual players in an exergame"

_PeerJ Computer Science, doi:10.7717/peerj-cs.92_

## Round 0.1 · original submission · Major Revisions

One reviewer recommends rejection, one recommends very major revision, and one recommends minor revision. A revision can be prepared but all of the reviewer comments need to be addressed. The paper as is, is not sound. The revision required is significant.

The claims made in the results need to be backed up with specific statistical testing. See comments by R1.

The implementation of “cooperation” needs to be discussed more critically. As R1 notes, the paper does not implement an intuitive case of cooperative performance. You should either give the condition a more transparent name or provide a better explanation of why you call it cooperative.

The issue of perceived cheating comes up throughout the paper but is not addressed directly. It is odd that motivation and enjoyment were operationalized in single questions, yet it is quite likely that these are the most important psychological dynamics in this research. See R3 and R2.

The overall logic of two studies, the progression of the research program, is not made clear. R1 and R3 both asked why this is presented as two studies and not one? This should be addressed.

Why were the studies carried out? Why were they structured as they are? What are the research questions? Why did you measure what you did and in the ways that you did? See R1 and R3

The paper should cite more of the HCI literature on exergaming, and be framed more clearly within that literature. The paper should more explicitly identify implications for the design of exergaming technology.

More minor point

There are inconsistencies in the experimental designs and reporting of results across the two studies. These should be explicitly acknowledged and justified. See comments by R1

Reviewer 1 ·

Basic reporting

This paper is generally well-written with minor typos (e.g., Line 94 -- "maybe" should be "may be", etc.). Due to the experimental nature of the study in a controlled laboratory setting, the Introduction section of the paper should be rewritten to include a "Research Questions" statement and the associated "Hypotheses" substantiated by prior research. The current literature review section should be separated into a standalone Related Work section. Also, it'd be great for the authors to state the primary contributions upfront at the end of the Introduction section to better guide the readers

Experimental design

The write up for the experimental design is generally clear but there exists a number of discrepancies in the reporting of study 1 and study 2 that can be improved. For example, in study 1, the participant assignments were "randomized" (line 121) whereas study 2 follows a "Latin Square" (line 317). Study 1 contains 3 conditions (e.g., control, ghost, and trainer; line 161-166) but study 2 contains also 3 conditions but is mentioned as "2" conditions (line 250-254) which is inconsistent with Table 3's caption. Study 2 did not measure RPE.

The paper does not provide a correlation table for study 1, but provides a correlation table for study 2 (though the addition of this table serves very little purpose due to the general lack of significant correlations). The authors could've done a dimension reduction to reduce SOQ, especially since none of the covariates correlate with any of the DVs. If anything, the paper should provide correlations of covariates and *all* DVs (also including calories, motivation, and enjoyment) to make the presentation more meaningful.

Significant relationships should be denoted in Table 1 and Table 2 (e.g., using superscripts a, b, c given a > b > c).

Validity of the findings

The paper contains several unsubstantiated claims. For example, statements in Line 232-233, 402-403, and 411-414 need to be justified with statistical analysis--need to provide N, posthoc comparisons, and clustering results (e.g., k-mean). Without stats, the claims are completely unsubstantiated.

The most significant issues with this paper is that the argument of competitive or cooperative is mischaracterized given the way the experimental conditions are implemented. Technically, paper experiments with 4 conditions--control (twice), ghost, basic trainer (inhuman agility), trainer-gamified (simulating more naturalistic AI-bot, which the authors classify as "competitive"), and trainer-unbeatable (supposed to simulate "cooperative" condition but fails to do so; see my comments later). By definition, "cooperation" is a situation in which the players work together toward a common goal, but the "cooperative" condition that is implemented in the experiment is one in which the AI-bot *always* wins without any cooperative element (see Figure 4 and line 398-401). This can also explain why enjoyment/motivation of competitive > control & cooperative (see Table 2). In the "cooperative" condition, the players always "lose" by definition. The way in which cooperative condition is implemented in this work is better described as a "tutorial" or an "assisted/cheat mode" in which optimal path is hinted. Again, there's no cooperative element in this implementation, which breaks the framing of competitive vs. cooperative framing of the paper.

The system should implement a feedback mechanism (e.g., with a distance or progress bar visual indicator indicating the gap in distance or audio feedback) so that the players can still know the AI's progress even if the AI falls behind (line 229).

The 2 studies can be treated as 1 and the analysis can be done across the 4 conditions stated above. It appears that the ghost condition may outperform any of the AI-bot conditions (comparing with self or with ghost conditions of others in similar profile; see Quantified Self literature). This may be a stronger argument that the current framing of the paper.

Additional comments

Future work needs to account for novel effect and halo effect that could be present in the described studies.

This paper requires *very* major revision due to the inherent issues in framing and the way in which the procedures are described and the statistical tests are analyzed/presented.

·

Basic reporting

This article is weel written. I just found one typo :

In the abstract : ... than playong on one's own with (a) the virtual trainer (delete a or the)

And an tiny error : line 110, authors mention three condition while there are only two.

Experimental design

The experimental design is well described. Results are clearly presented

Validity of the findings

Some choices could be explain :
Are 5 minutes enough to separate two sessions. I'm not expert in sport but it seems that this duration is short and even the control condition can decrease the fitness and then the performances of users. In the same line, even if different orders are proposed to different users, it could be interesting to test a possible order effect on the data.

Why do you evaluate subjective feeling with only one question. Generally, a questionnary includes some redundant questions and an evaluation of the coherency of the different answers to these question is evaluated.

I would like to know which condition is used to record the ghost (is the user already with a trainer or not ?). This point could modify the behavior of the ghost and then, the results. Whatever the condition is, it is important to mention it and also to discuss the impact of the recording of the ghost on the results.

To finish, it is always more easy to compare conditions. However, independant conditions measurment can be suprising. It could be interessesting that the authors give their opinion on this point. What were the results if they had test only one condition without comparaison ?

Additional comments

Interesting work. I hope that you can take into account some of my propositions.

Reviewer 3 ·

Basic reporting

This is a promising paper reporting two cross-sectional studies investigated the effects of competition and cooperation on exercise performance in an immersive Virtual Reality (VR) cycle exergame. The results, as claimed by the authors, demonstrate that a competitive experience in exergaming is an effective tool to elicit higher levels of exercise from the player, and can be achieved through virtual substitutes (e.g., the ghost or virtual trainer) for another human player.

The introduction and motivation to the study can be better argued by reviewing more related studies and explaining the gaps in current research. It is unclear why the 2 studies need to manipulate the social factors of competition and cooperation, and encourage the player to maintain an appropriate level of exercise. Why are the study conditions considered in the 2 studies?

The paper is generally presented in an easy-to-read manner. The authors may like to check out in-text citation reporting format. Attention should also be paid to the use of tenses when reporting studies conducted and review of literature, for example.

Attached are some comments made on the manuscript.

Experimental design

Broadly, I think the study design can be explained in more details:
- Why is the exergame selected?
- Why are the 3 conditions tested? What hypotheses are formulated?
- Explain the sampling profiles. Two studies had more males. Could the study findings be biased?
- Why are the measurement items selected? In the case of the "enjoyment" and "motivation" constructs, it is a one-item measurement. Is this sufficient?

Validity of the findings

As highlighted above, the experimental designs may lead to biases in the findings.

It would have been better to combine the 2 studies into a single study and the same subjects be tested under the conditions, so the findings can be more accurately interpreted.

Additional comments

This is an interesting paper and I think the authors are commended on embarking the 2 studies. If it could be carried out again, it is advised that the authors think about the hypotheses they would like to test. The recruitment of the subjects should also be more representative of the population group they intend to study and indicate the reasons for selecting this group to study.

---

## Round 0.2 · accepted · Accept

The reviewer suggested an approach to formatting Table 1 and 2 to make significance of differences clearer. Please consider this minor change while in the production phase.

Reviewer 1 ·

Basic reporting

See below.

Experimental design

See below.

Validity of the findings

See below.

Additional comments

I'm satisfied with the revised manuscript. I do wish that the authors would attempt to experiment with different types of competitive and cooperative elements in future studies, but that's beyond the scope of this paper. In its current form, I do think that the paper provides novel contribution to the exergame literature.

One minor suggestion: when reporting significant differences using superscripts in Table 1 and Table 2, superscripts should be used when a relationship a > b > c exists. In other words, superscripts should be omitted when no difference is present. The convention of using superscripts to denote a > b is similar to using asterisks to denote statistically significant comparisons. If there's no difference and a superscript "a" is denoted across all means, then it makes it extremely difficult to read.